# Anti-SARS-CoV-2 Activity of Rhamnan Sulfate from *Monostroma nitidum*

**DOI:** 10.3390/md19120685

**Published:** 2021-11-30

**Authors:** Yuefan Song, Peng He, Andre L. Rodrigues, Payel Datta, Ritesh Tandon, John T. Bates, Michael A. Bierdeman, Chen Chen, Jonathan Dordick, Fuming Zhang, Robert J. Linhardt

**Affiliations:** 1Departments of Chemistry and Chemical Biology, Center for Biotechnology and Interdisciplinary Studies, Rensselaer Polytechnic Institute, Troy, NY 12180, USA; songy11@rpi.edu (Y.S.); hep3@rpi.edu (P.H.); payel.datta@acphs.edu (P.D.); 2Department of Chemical and Biological Engineering, Center for Biotechnology and Interdisciplinary Studies, Rensselaer Polytechnic Institute, Troy, NY 12180, USA; lopesa@rpi.edu (A.L.R.); dordick@rpi.edu (J.D.); 3Department of Microbiology and Immunology, University of Mississippi Medical Center, Jackson, MS 39216, USA; rtandon@umc.edu (R.T.); jtbates@umc.edu (J.T.B.); 4Department of Medicine, University of Mississippi Medical Center, Jackson, MS 39216, USA; mbierdeman@umc.edu; 5Calroy Health Sciences, LLC., Scottsdale, AZ 85260, USA; chenchen99@gmail.com; 6Departments of Biological Science and Biomedical Engineering, Center for Biotechnology and Interdisciplinary Studies, Rensselaer Polytechnic Institute, Troy, NY 12180, USA

**Keywords:** SARS-CoV-2, rhamnan sulfate, heparin, surface plasmon resonance

## Abstract

The COVID-19 pandemic is a major human health concern. The pathogen responsible for COVID-19, severe acute respiratory syndrome coronavirus 2 (SARS-CoV-2), invades its host through the interaction of its spike (S) protein with a host cell receptor, angiotensin-converting enzyme 2 (ACE2). In addition to ACE2, heparan sulfate (HS) on the surface of host cells also plays a significant role as a co-receptor. Our previous studies demonstrated that sulfated glycans, such as heparin and fucoidans, show anti-COVID-19 activities. In the current study, rhamnan sulfate (RS), a polysaccharide with a rhamnose backbone from a green seaweed, *Monostroma nitidum*, was evaluated for binding to the S-protein from SARS-CoV-2 and inhibition of viral infectivity in vitro. The structural characteristics of RS were investigated by determining its monosaccharide composition and performing two-dimensional nuclear magnetic resonance. RS inhibition of the interaction of heparin, a highly sulfated HS, with the SARS-CoV-2 spike protein (from wild type and different mutant variants) was studied using surface plasmon resonance (SPR). In competitive binding studies, the IC_50_ of RS against the S-protein receptor binding domain (RBD) binding to immobilized heparin was 1.6 ng/mL, which is much lower than the IC_50_ for heparin (~750 ng/mL). RS showed stronger inhibition than heparin on the S-protein RBD or pseudoviral particles binding to immobilized heparin. Finally, in an in vitro cell-based assay, RS showed strong antiviral activities against wild type SARS-CoV-2 and the delta variant.

## 1. Introduction

Severe acute respiratory syndrome coronavirus 2 (SARS-CoV-2), a single-stranded positive RNA virus bearing a membrane envelope, is the pathogen that causes COVID-19. The invasion of SARS-CoV-2 is achieved by the interaction of its spike protein (S-protein) with angiotensin-converting enzyme 2 (ACE2), which acts as the major receptor for the S-protein, leading to viral entry into susceptible cells. In addition to this major receptor, there are also co-receptors/cofactors that play significant roles in the invasion process [1]. One of these co-receptors, heparan sulfate (HS), was revealed to be required for SARS-CoV-2 infection, as it facilitates SARS-CoV-2 S-protein binding to ACE2 and assists viral entry [2,3]. The S-protein interacts with both cellular HS and ACE2 through its receptor-binding domain (RBD). It is believed that no dissociation between HS and the RBD is required for binding to ACE2 [4]. Analogs of HS, heparin, and non-anticoagulant derivatives block SARS-CoV-2 binding and infection. Heparin, a highly sulfated version of HS, shows stronger inhibition of S-protein binding to cells than HS [5]. SPR competition assays revealed that heparin has a higher binding preference for the SARS-CoV-2 S-protein than various desulfated heparin derivatives and other glycosaminoglycans (GAGs), indicating that the binding of GAGs to the S-protein is greatly influenced by their degree of sulfation [6]. These data suggest that the sulfated glycans, including the natural products of a variety of organisms, may represent promising therapeutics for the prevention or treatment of COVID-19 infection.

Natural products from marine organisms have served as a highly diverse resource for discovering new compounds with potential pharmacologic potential. Among various marine bioresources, algae are considered an outstanding source of sulfated glycans, particularly brown, red, and green algae. Different kinds of algae contain different types of sulfated glycans, such as fucoidan, a fucose-containing sulfated glycan found in brown algae, and agar and carrageenan found in red algae. In green algae, ulvans are the principal cell wall matrix polysaccharides of the genus *Ulva* [7], and rhamnan sulfates (RS) are found in the genus *Monostroma* [8].

Marine sulfated polysaccharides often show anticoagulant activity like that exhibited by heparin. Sulfated glycans from green algae, typically consisting of rhamnose, also show anticoagulant activity [8,9,10,11,12,13,14,15]. RS is also bioactive, exhibiting antioxidative, anti-hyaluronidase, antitumor, anti-obesity, anti-hypercholesterolemic, anti-hyperglycemic, and anti-diabetes activity [16,17,18]. RS from *Monostroma* has activity against the measles virus, mumps virus, influenza A virus, human coronavirus 229E, human immunodeficiency virus (HIV), human cytomegalovirus (HCMV), enterovirus 71, and herpes simplex virus types 1 and 2 (HSV-I and -II); however, it does not act on the adenovirus, poliovirus, coxsackie virus, and rhinovirus [16,19,20,21,22].

Combating SARS-CoV-2 has become an unprecedented priority, as different variants have resulted in prevailing surges of COVID-19. In the current study, we recovered the RS from the green algae *Monostroma nitidum*, analyzed the RS composition and structure, and evaluated its binding to the S-protein RBD from the wild type and different mutant variants of SARS-CoV-2 using SPR competition experiments. Based on our prior work on the binding of the S-protein to heparin and its derivatives, unfractionated heparin generally showed better performance for binding to the S-protein, S-protein RBD, or viral particles than its desulfated or short-chain derivatives [6,23,24,25]. Therefore, we used unfractionated heparin in our SPR experiments as both a chip surface molecule and/or a competitive S-protein binder in buffer, and tested heparin binding to different S-protein variants to provide a positive control for RS from *M. nitidum.* Finally, we confirmed the neutralization capacity of RS for the pseudotype virus in a cell culture assay. These data suggest the RS from *M. nitidum* may be a promising candidate in COVID-19 treatment or prevention, and that it is effective against variant S-proteins of SARS-CoV-2 mutants.

## 2. Results and Discussion

### 2.1. Preparation and Compositional Analysis of RS

A total of 2.0 g of crude extract was obtained from 8.0 g of *Monostroma nitidum* powder. Further purification using an anion exchange Q-column recovered 1.24 g of RS from the crude extract, giving a yield of 62% (*w*/*w*) based on crude extract, or 15% (*w*/*w*) based on dried powder. The isolated RS contained 59 ± 9% (*w*/*w*) carbohydrate and 31 ± 4% (*w*/*w*) sulfate. The monosaccharide composition of RS, as determined by chromatographic analysis, was determined to be (molar ratio, expressed as relative to rhamnose) rhamnose:1, mannose:0.02, glucuronic acid:0.06, glucose:0.10, galactose:0.02, xylose:0.08 (Figure 1). Unsurprisingly, rhamnose is the predominant monosaccharide, which accounts for 78% (molar percentage) of the total sugar. Based on gel permeation chromatography, the isolated RS had an average molecular weight of 290 kDa and a number average molecular weight of 53 kDa (Figure 2), giving a polydispersity index of approximately 5.5. Based on the above data, RS is a highly sulfated, rhamnose-predominant polysaccharide fraction with a high average molecular weight and a wide molecular weight distribution. It is a major constituent in *M. nitidum,* considering its high yield from the dry powder.

### 2.2. 2D-NMR Analysis of RS

The elucidation of RS structure from 2D NMR was performed by cross-checking the signals of HSQC (Figure 3) and TOCSY spectra (Appendix A). The RS backbone mainly consists of →3)-α-L-Rha*p*-(1→, →2)-α-L-Rha*p*-(1→, and →2, 3)-α-L-Rha*p*-(1→units. The C4 hydroxyl groups of RS are largely sulfated, which can be concluded from the intense signal in the C2-C5 region of the HSQC spectrum (76–80 (f1)/4.2–4.4 (f2) ppm, Figure 3B). The non-reducing end of RS is heavily sulfated at C4 and/or C3 (Figure 3A). RS also contains rhamnose units that are sulfated at both C2 and C4 hydroxyl groups. The C1 of the reducing end was assigned at 92–94 ppm. However, the signal at 96–98 (f1)/5.6–5.8 (f2) ppm in HSQC suggests that some C1 hydroxyl group at the reducing end is sulfated. In addition, minor signals of →4)-β-D-GlcA*p*-(1→ and →6)-β-D-Glc*p*-(1→ were also assigned in the anomeric region.

### 2.3. Interaction between SARS-CoV-2 S-Protein RBD and Heparin

SPR was employed to measure the binding kinetics and affinity of the SARS-CoV-2 S-Protein RBD interaction (wild type (wt) and variants) with heparin using a sensor chip with immobilized heparin (Figure 4 and Figure 5). Binding kinetic parameters (k_a_ and k_d_) and affinity (K_D_) were calculated (Table 1) from sensorgrams globally fitted to a 1:1 Langmuir model using T200 Evaluation software. Most of the binding kinetics and affinity of different variants for the S-Protein RBD are comparable to the wild type version, except the N501Y which shows higher affinity, and L452R with lower affinity to heparin.

### 2.4. IC_50_ Measurement on the Inhibition of S-Protein Binding to Heparin by RS Using Solution Competition SPR

To examine the ability of RS to inhibit the interaction between heparin (on surface) with the wt S-protein, solution competition dose response analysis was performed between surface immobilized heparin and soluble RS to calculate IC_50_ values (Figure 6). S-protein was pre-mixed with different concentrations of RS or heparin before injection into the heparin chip. Once the active binding sites on the S-protein were occupied by glycan in solution, its binding to the surface-immobilized heparin decreased, resulting in a reduction of signal in a concentration-dependent fashion. The IC_50_ values were calculated from the plots of S-protein binding signal (normalized) vs. glycan concentration in solution. The IC_50_ value of RS was 1.6 ng/mL, which is much lower than the positive control heparin (IC_50_ ~750 ng/mL).

### 2.5. Inhibition of S-Protein RBD Variants Binding to Heparin by RS Using Solution Competition SPR

Solution/surface competition experiments were also performed using SPR to examine the ability of RS to inhibit the interaction of S-Protein RBD variants with heparin (Figure 7). Using the same concentration of RS and heparin (5ng/mL), RS showed stronger inhibition in the S-protein variants E484Q and L452R+E484Q, and comparable inhibition in T478K to that of heparin.

### 2.6. SPR Solution Competition Study on the Inhibition by RS of Heparin Interaction with Pseudovirus Particles

Solution/surface competition experiments were performed to quantify the inhibition by RS of various pseudovirus particle (wild type and delta variant)–heparin interactions (Figure 8). RS showed stronger inhibition in the wild type and comparable inhibition in the delta variant pseudovirus particles binding to surface heparin than the heparin in solution.

### 2.7. In Vitro SARS-CoV-2 Pseudotyped Virus Neutralization

Glycans such as heparin have been shown to inhibit viral infection by interacting with the SARS-CoV-2 spike protein [26]. As such, we investigated the ability of RS to inhibit viral entry via a cell-based neutralization assay of SARS-CoV-2 pseudoviral particles. HEK293T cells that stably express the ACE-2 receptor (HEK293T-ACE2) were used. Six different concentrations were tested at a 1:10 viral dilution, thus enabling the determination of an IC_50_ value for viral inhibition based on the expression of EGFP as a marker for functional viral entry. Briefly, RS was incubated with pseudovirus particles for 1 h at 37 °C, after which the mixture was added to the HEK293T-ACE2 cells and incubated for 4 h. These incubation steps were performed under xeno-free conditions, as serum often contains growth factors that interact with polysaccharides and thus interfere with their interaction with the spike protein. After the 4 h incubation, there was a media exchange with serum to sustain cell growth for 48 h, after which the plates were assayed for expression of EGFP. The results of the neutralization experiment are shown in Figure 9. The *y*-axis represents the percentage of maximum infectivity that could be obtained for the experiment. For each concentration, the percentage of infected cells was normalized to the percent infected cells relative to the control (no RS and 1:10 viral dilution). As can be seen in Figure 9, the lowest dilution at an RS concentration of 1 μg/mL provided >80% viral entry inhibition for both the wild type and delta variant, thus leading to an IC_50_ value of 2.39 and 1.66 μg/mL, respectively.

The algal-derived sulfated glycan has been proven to be effective in binding the S-protein of SARS-CoV-2, which could disturb the S-protein interacting with cell surface heparan sulfate leading to viral attachment. Besides inhibiting viral attachment, algal-derived sulfated glycans also inhibit virus penetration, exteriorization, uncoating, transcription, and translation processes [27]. Indeed, in previous work, the fucoidan from brown alga and the carrageenan from red alga possess antiviral activity [26,28,29,30]. In this research, we found that RS from *M. nitidum* is as competitive as the fucoidans and iota-carrageenan in its potential to combat COVID-19 [26,28], which is reflected by its high capacity in binding the S-protein and neutralizing the pseudotyped virus in vitro.

## 3. Materials and Methods

### 3.1. Biological Materials and Reagents

Dried *Monostroma nitidum* powder was provided from Calroy Health Sciences, LLC. SARS-CoV-2 S-Protein RBD (wild type, wt) and N501Y were expressed in Expi293F cells provided by the Bates lab, University of Mississippi Medical Center. SARS-CoV-2 S-Protein RBD mutants (related to delta variants of SARS-CoV-2) were purchased from Sino Biological US Inc. (Wayne, PA). SARS-CoV-2 pseudoviral particles (wt and delta variant) were prepared as previously described [25]. Sensor SA chips were from Cytiva (Uppsala, Sweden). Monosaccharide standards for L-rhamnose monohydrate, D-mannose, D-glucosamine hydrochloride, D-glucuronic acid, D-glucose, D-galactose, L-fucose, D-xylose, and the dextrans used in gel permeation chromatography (GPC) were from Millipore Sigma Co. (Bedford, MA, USA). Vivapure^®®^ D Maxi H columns were purchased from Sartorius Stedim North America (Bohemia, NY, USA). All the reagents used in the HPLC experiment were of HPLC grade.

### 3.2. Preparation of RS

*M. nitidum* powder (8 g) was added to 160 mL distilled water and placed at 4 ℃ overnight. The solution was then incubated at 70 ℃, stirred for 1 h, and then centrifuged at 4000 rpm for 40 min. The supernatant was collected, and the residue was extracted with 160 mL distilled water again as described above. The supernatants were combined, dialyzed for 48 h, and lyophilized to provide a crude extract. The crude extract was further dissolved in distilled water and loaded into a Vivapure^®^ D Maxi H column at an amount of 20 mg/column. The neutral impurities were removed by washing the column with distilled water. The anionic polysaccharide was collected by washing the column with 2 M NaCl, and then dialyzed and lyophilized before use.

### 3.3. Compositional Analysis of RS

The total sugar content of RS was determined using the phenol–H_2_SO_4_ reaction [31] and detected at 490 nm. L-rhamnose:D-galactose (4:1) was used as a standard in the total sugar content assay. The sulfate content of the RS was determined via the BaCl_2_-gelatin turbidimetric method. In brief, 2 mg/mL RS was hydrolyzed in 2 M HCl at 70 ℃ for 24 h, cooled down to room temperature, and diluted to 0.5 mg/mL with distilled water. Eighty microliters of the above solution were mixed with 1% gelatin (*w*/*v*) and 20% BaCl_2_ (*w*/*v*) at a volume ratio of 1:1:1 in a 96-well plate. The standard curve was constructed using 0–80 µL 0.05% K_2_SO_4_ (*w*/*v*) in 0.5 M HCl instead of RS hydrolyzate, and distilled water was supplemented to the K_2_SO_4_ solution to provide a total volume of 80 µL. Sulfate content was determined using a 96-well plate and measuring absorbance at 600 nm using a Biotek plate reader (Winooski, VT, USA), and the RS sulfate content was calculated by reference to the standard curve. The monosaccharide composition was determined using a high-performance liquid chromatography (HPLC) method [32] with minor modification. The RS was hydrolyzed with 4 M HCl at 70 °C for 24 h and left overnight at 4 °C. The acidic solution was then neutralized using 4 M NaOH and subjected to derivatization with 3-methyl-1-phenyl-2-pyrazoline-5-one in NaOH solution. The reaction was terminated by neutralization with acetic acid, and the 3-methyl-1-phenyl-2-pyrazoline-5-one residue was removed by extraction with chloroform. The monosaccharide reference standards for rhamnose, mannose, glucosamine, glucuronic acid, glucose, galactose, fucose, and xylose were also subjected to the same pre-column derivatization procedure. The HPLC analysis was performed using a Shimadzu LC-40D system and an Agilent Poroshell 120 EC-C18 column (2.7 µm, 2.1 × 100 mm). The mobile phase was aqueous 0.05 M KH_2_PO_4_ (pH 6.9) with 15% acetonitrile as solvent A and 0.05 M KH_2_PO_4_ (pH 6.9) with 40% acetonitrile as solvent B. A gradient was performed as follows: 0 min, 90% A→ 7.5 min, 90% A → 30 min, 50% A. The flow rate was 250 µL/min. Five microliters of sample were injected into the HPLC and detection was performed at 254 nm. The peak areas were calculated with LabSolution software. The molecular weight of RS was estimated by GPC using a Shimadzu LC-10A HPLC and detected via refractive index. A 10 × 300 mm column packed with Sephacryl S-500 HR resin was used. The mobile phase was 0.1 M Tris-HCl (pH 7.5) at a flow rate of 0.5 mL/min. The molecular weight was assessed by reference to a calibration curve made by dextran standards (the weight average molecular weights are 2000 k, 670 k, 410 k, 150 k, and 50 k Da, respectively).

### 3.4. 2D NMR Analysis

2D NMR was performed using a Bruker SB 800 MHz NMR Spectrometer (Billerica, MA, USA). The RS sample was dissolved in D_2_O, placed at 4 ℃ overnight, and lyophilized. This D_2_O replacement procedure was repeated twice before the RS sample was finally dissolved in D_2_O at a concentration of 30 mg/mL and examined at 298K for HSQC and TOCSY spectra.

### 3.5. Measurement of Interaction between Heparin and S-Proteins RBD Using SPR

The interaction of heparin and S-protein was measured by surface plasmon resonance (SPR) using a BIAcore T200 and T200 evaluation software (Uppsala, Sweden). Biotinylated heparin was prepared and immobilized to a streptavidin (SA) chip based on the manufacturer’s protocol. The successful immobilization of heparin was confirmed by the observation of a ~200 resonance unit (RU) increase in the sensor chip. The S-protein RBD (wt and variants) samples were diluted in HBS-EP buffer (0.01 M HEPES, 0.15 M NaCl, 3 mM EDTA, 0.005% surfactant P20, pH 7.4). Different dilutions of protein samples were injected at a flow rate of 30 µL/min. At the end of the sample injection, the same buffer was flowed over the sensor surface to facilitate dissociation. After a 3 min dissociation time, the sensor surface was regenerated by injection with 30 µL of 2 M NaCl to fully regenerate the surface. The response was monitored as a function of time (sensorgram) at 25 ℃.

### 3.6. SPR Solution Competition Study of RS

Solution competition between surface heparin and RS in buffer was performed using SPR [33]. In brief, S-Protein RBD (250 nM) and SARS-CoV-2 pseudoviral particles (10 times dilution with HBS-EP buffer) mixed with RS in HBS-EP buffer were injected over the heparin chip at a flow rate of 30 µL/min. Once the active binding sites on the protein molecules are occupied by RS in solution, the binding of S-Protein RBD/pseudoviral particles to the surface-immobilized heparin should decrease, resulting in a reduction of signal. After each run, dissociation and regeneration were performed as described above. For each set of competition experiments on SPR, a control experiment (protein only) was performed to ensure that the chip surface was completely regenerated and that the results obtained between runs were comparable. The same protein samples were also mixed with heparin in HBS-EP buffer and were tested to serve as a positive control.

### 3.7. In Vitro SARS-CoV-2 Pseudotyped Virus Neutralization Assay

*ACE-2 stable cell line generation:* Lentiviral particles containing the ACE2-Puro construct were produced by transfecting in 12.3 μg psPAX2 (Addgene # 12260), 2.5 μg pMD2g (#12259), and 14.7 μg pLenti-hACE2-Puro into HEK293T cells using Lipofectamine 2000 according to the manufacturer’s instructions. The plasmids, psPAX2 and pMD2g, were a gift from Didier Trono (École Polytechnique Fédérale de Lausanne, Switzerland). A medium exchange was carried out 24 h after transfection and 5 mM sodium butyrate (Millipore Sigma, Burlington, MA, USA) was added to the cells in fresh medium. The supernatant from HEK293T cells carrying the lentiviral particles were harvested at 48 h and 72 h. The supernatants were pooled and concentrated using Lenti-X-Concentrator (Takara Bio, Shiga, Japan) according to the manufacturer’s instructions. The concentrated lentiviral particles carrying ACE2-Puro were delivered to HEK293T cells in 6-well tissue culture treated plates. After 48 h, 4 μg/mL of puromycin was added to DMEM+10% FBS and a medium exchange was carried out. The cells were passaged to a T-25 flask and maintained in selection pressure (4 μg/mL puromycin) to remove cells lacking the ACE2-Puro construct.

*Production of Spike Pseudotyped Viral Particles:* HEK293T cells were seeded in two T175 flasks and cultured in DMEM + 10%FBS. At 70–80% confluence, the cells were transfected using Lipofectamine 2000. For production of wild type and delta spike pseudotyped particles, the cells were transfected with 26 μg of psPAX2, 26 μg of pLV-EGFP (a gift from Pantelis Tsoulfas (University of Miami, Florida), Add gene plasmid # 36083), and 8.7 μg of pHDM-SARS-CoV-2-S (BEI Resources #NR52514) per flask. A medium exchange was performed 24 h after transfection with the addition of 5 mM sodium butyrate (Millipore Sigma, Burlington, MA, USA). The harvest supernatant was collected at 48 h and 72 h and concentrated using Lenti-X-Concentrator according to the manufacturer’s instructions. The resuspended viral samples were stored at −80 ℃ until use.

*SARS-CoV-2 Pseudotyped Virus Neutralization Assay:* Six different concentrations of RS were prepared at a 10-fold serial dilution from 1000 to 0.01 μg/mL in DMEM + 1% PenStrep and no FBS. Viral samples were then added at a 1:10 dilution and the mixtures were incubated at 37 °C for 1 h. The samples were then added to HEK293T-ACE2 cells, plated in 96-well plates at 15,000 cells/well, and incubated for 4 h at 37 °C. Afterwards, a media change was performed with DMEM + 1% PenStrep + 10% FBS. The cells were cultured for an additional 48 h and were then stained with 5 μg/mL of Hoechst 33,342 and imaged using Cellomics Arrayscan XTI. The infection efficiency was then calculated using the Target Activation Bioapplication. The results of the experiment represent the percent of maximum infectivity that could be obtained for the experiment. This was done by normalizing the percent infected value for each sample by the percent infected value at the 1:10 dilution and 0 μg/mL of compound.

## 4. Conclusions

We explored the anti-SARS-CoV-2 activity of a RS fraction from the green algae *M. nitidum*. RS is a highly sulfated, rhamnose-predominant, high-molecular-weight glycan fraction. The RS backbone mainly consists of →3)-α-L-Rhap-(1→, →2)-α-L-Rhap-(1→, and →2, 3)-α-L-Rhap-(1→ units, with the C4 hydroxyl groups largely sulfated. SPR confirmed the affinity of chip-surface heparin with the S-protein RBD from the wild type SARS-CoV-2 and a variety of variants. In a competition SPR assay, the RS in solution showed impressive activity of inhibiting chip-surface heparin binding with the wild type S-protein RBD (with a measured IC_50_ = 1.6 ng/mL), while for heparin in solution, which was used as the control, the IC_50_ = ~750 ng/mL. RS also showed a higher capacity in binding the S-protein RBD from a variety of variants, and the pseudovirus particles of the wild type and delta variant, compared to that of heparin at the same concentration (5 ng/mL). Finally, we confirmed the neutralizing effect of RS on the SARS-CoV-2 pseudotyped virus in vitro, with an IC_50_ value of 2.39 and 1.66 μg/mL to the wt and delta variant, respectively. To develop RS as a therapeutic and/or preventative antiviral drug, more studies are proposed in future investigations, including the structure activity relationship (SAR), bioavailability, and antiviral activity of low molecular weight RS and a toxicity analysis of RS.

## Figures and Tables

**Figure 1 marinedrugs-19-00685-f001:**
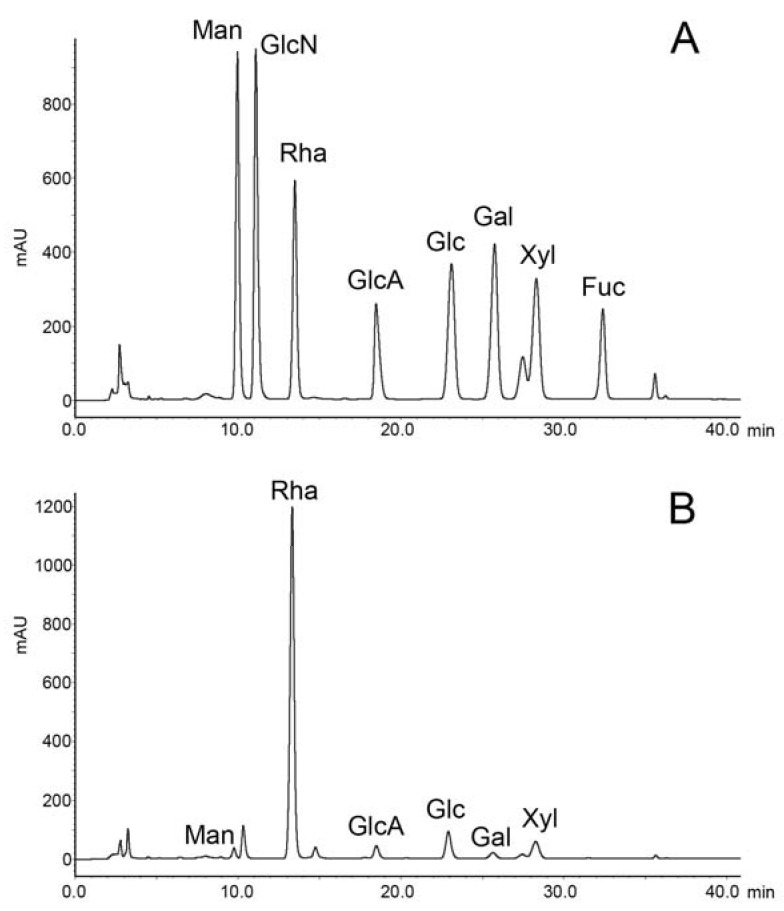
Chromatography of monosaccharide analysis. (**A**) Monosaccharide standards. (**B**) Rhamnan sulfate from the *M. nitidum*.

**Figure 2 marinedrugs-19-00685-f002:**
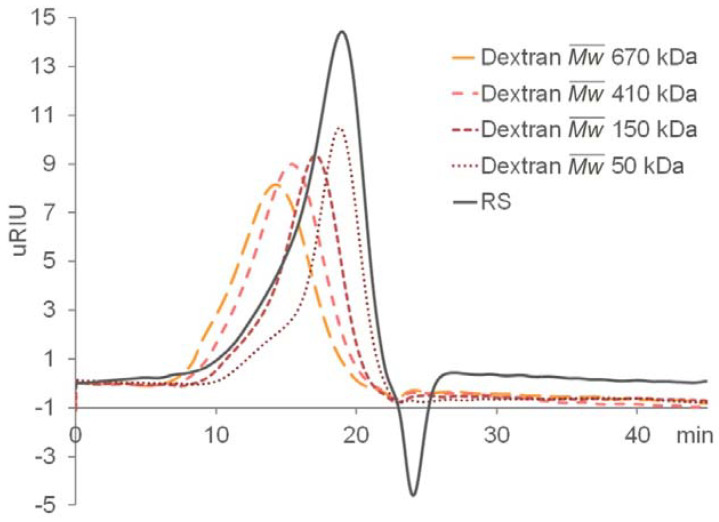
Gel permeation chromatography (GPC) of RS and the dextran standards.

**Figure 3 marinedrugs-19-00685-f003:**
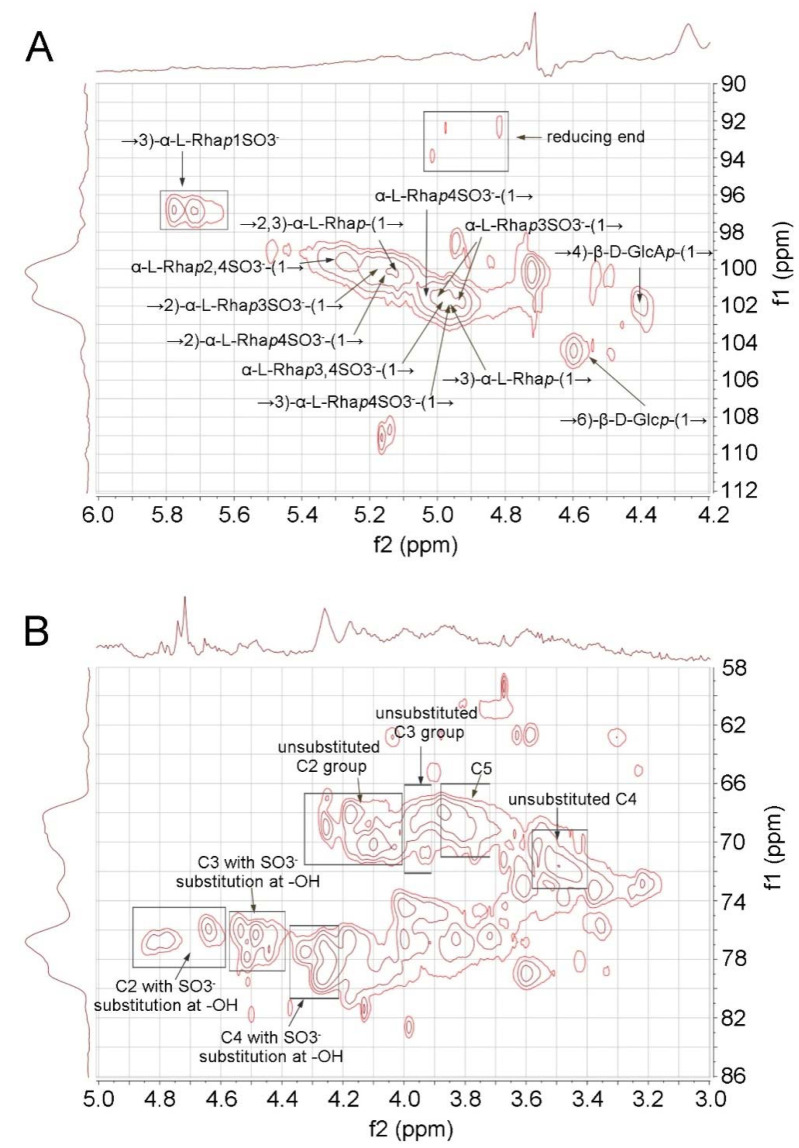
HSQC spectrum of RS. (**A**) Anomeric region. (**B**) C2-C5 region.

**Figure 4 marinedrugs-19-00685-f004:**
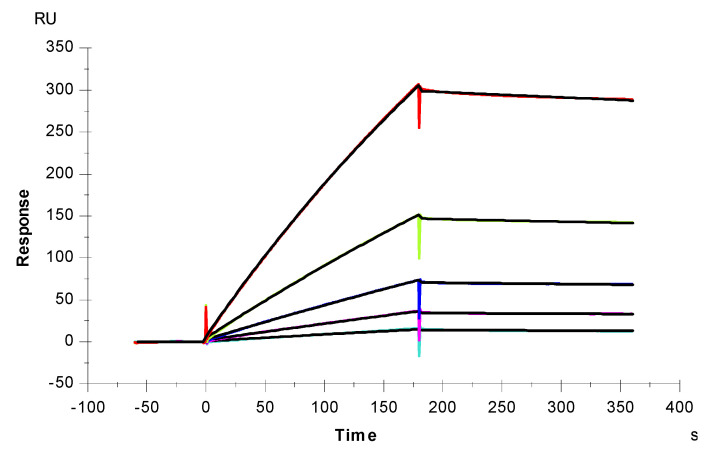
SPR sensorgrams of S-Protein RBD (wt) interaction with heparin. Concentration of S-Protein RBD (from top to bottom): 1000, 500, 250, 125, and 63 nM, respectively. The black curves are the fitting curves using models from T200 Evaluate software.

**Figure 5 marinedrugs-19-00685-f005:**
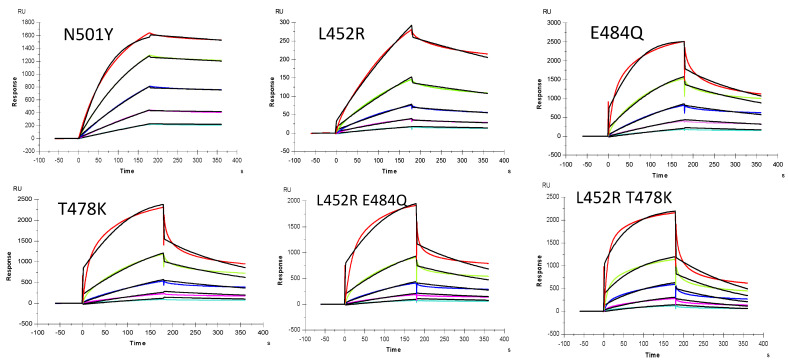
SPR sensorgrams of S-Protein RBD mutants’ interaction with heparin. Concentration of S-Protein mutants (from top to bottom): 1000, 500, 250, 125, and 63 nM, respectively. The black curves are the fitting curves using models from T200 Evaluate software.

**Figure 6 marinedrugs-19-00685-f006:**
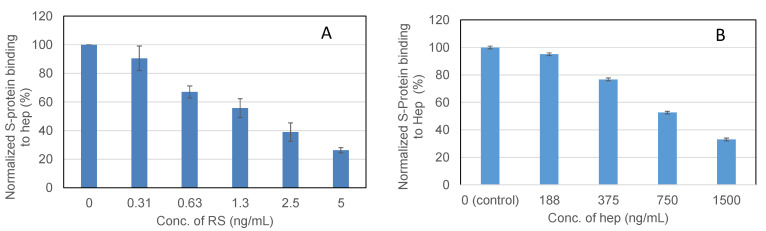
IC_50_ measurement on the inhibition by RS of S-Protein binding to heparin using solution competition SPR. Measured IC_50_ = 1.6 ng/mL for RS (**A**); IC_50_ = ~750 ng/mL for heparin (**B**). Error bars represent standard deviations from triplicated tests.

**Figure 7 marinedrugs-19-00685-f007:**
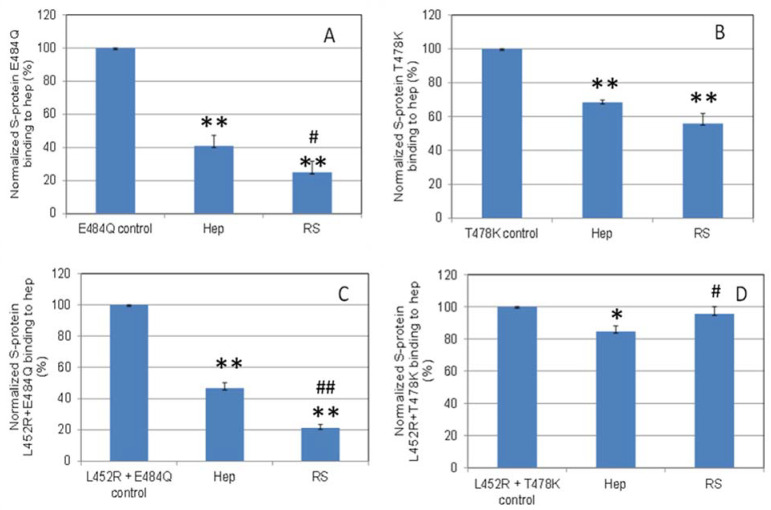
Bar graphs (based on triplicate experiments with standard deviation) of normalized S-protein mutants: (**A**) E484Q; (**B**) T478K; (**C**) L452R+E484Q; (**D**) L452R+T478K binding to surface heparin by inhibition with RS or heparin at concentration of 5 ng/mL in solution. Statistical analysis was performed using two-tailed Student’s t test to compare two different independent groups: * *p* < 0.05 compared to the control, ** *p* < 0.01 compared to the control; # *p* < 0.05 compared to the heparin, ## *p* < 0.01 compared to the heparin.

**Figure 8 marinedrugs-19-00685-f008:**
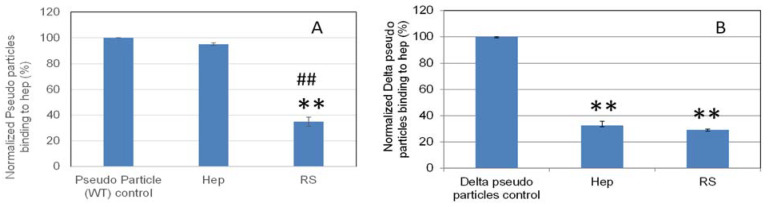
Bar graphs (based on triplicate experiments with standard deviation) of normalized pseudovirus particles binding to surface heparin by inhibition with RS or heparin. (**A**) Wild type pseudovirus particles inhibited by RS or heparin at concentration of 5 ng/mL in solution. (**B**) Delta variant pseudovirus particles inhibited by RS or heparin at concentration of 5 µg/mL in solution. Statistical analysis was performed using two-tailed Student’s t test to compare two different independent groups: ** *p* < 0.01 compared to the control; ## *p* < 0.01 compared to the heparin.

**Figure 9 marinedrugs-19-00685-f009:**
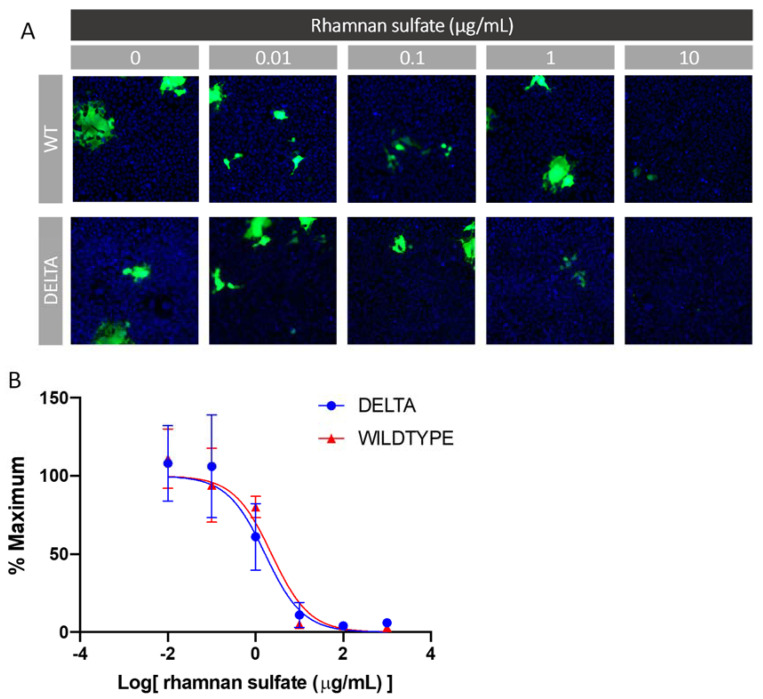
In vitro SARS-CoV-2 pseudotyped virus (wt and delta variant) neutralization assay. (**A**) Representative fluorescence microscopy of different concentrations of RS inhibitor assay. (**B**) IC_50_ curves of RS inhibiting SARS-CoV-2 pseudotyped virus (wt and delta variant).

**Table 1 marinedrugs-19-00685-t001:** Summary of kinetic data of heparin and SARS-CoV-2 S-protein RBD (wt and mutants) interactions *.

Interaction	k_a_ (1/*MS*)	k_d_ (1/*S*)	*K_D_* (*M*)
SARS-CoV-2 S-protein RBD wt	2144 (± 19)	2.2 × 10^−4^ (±2 × 10^−6^)	1.1 × 10^−7^
SARS-CoV-2 S-protein RBD N501Y	2.2 × 10^4^ (±66)	3.1 × 10^−4^ (±2.7 × 10^−6^)	1.4 × 10^−8^
SARS-CoV-2 S-protein RBD E484Q	3.7 × 10^4^ (±640)	5.0 × 10^−3^ (±7.9 × 10^−5^)	1.3 × 10^−7^
SARS-CoV-2 S-protein RBD L452R	1036 (±11)	1.3 × 10^−3^ (±4.1 × 10^−6^)	1.2 × 10^−6^
SARS-CoV-2 S-protein RBD T478K	3.2 × 10^4^ (±1.1 × 10^3^)	7.9 × 10^−3^(±2.2 × 10^−4^)	2.5 × 10^−7^
SARS-CoV-2 S-protein RBD L452R+E484Q	2.0 × 10^4^ (±800)	5.4 × 10^−3^ (±1.6 × 10^−4^)	2.7 × 10^−7^
SARS-CoV-2 S-protein RBD L452R+T478K	1.2 × 10^4^ (±89)	4.8 × 10^−3^ (±2.7 × 10^−5^)	4.1 × 10^−7^

* The data with (±) in parentheses are the standard deviations (SD) from global fitting of five injections.

## Data Availability

Not application.

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
