# Peer review of "Anti-SARS-CoV-2 Activity of Rhamnan Sulfate from Monostroma nitidum"

_marinedrugs, 2021, doi:10.3390/md19120685_

Round 1

Reviewer 1 Report

The manuscript entitled “ Anti-SARS-CoV-2 activity of rhamnan sulfate from Monostroma nitidum” by Song et al, aimed to evaluate the anti-SARS-CoV-2 activity of a polysaccharide rhamnan sulfate (RS). Since the COVID-19 pandemic continues to affect the wellbeing and life-style of people all over the world it is important to search a viral drug that can treat or prevent the COVID-19 infection.

The work is well organized and well written.

Minor concerns:

  • The references start with the 20,21 (line 45)
  • Statistical analyses must be inserted (figure 6, 7, 8, etc)

Author Response

The manuscript entitled “ Anti-SARS-CoV-2 activity of rhamnan sulfate from Monostroma nitidum” by Song et al, aimed to evaluate the anti-SARS-CoV-2 activity of a polysaccharide rhamnan sulfate (RS). Since the COVID-19 pandemic continues to affect the wellbeing and life-style of people all over the world it is important to search a viral drug that can treat or prevent the COVID-19 infection.  The work is well organized and well written.

Response: We thank reviewer #1 for the positive and constructive comments on our manuscript. 

Comments:

  1. The references start with the 20,21 (line 45)

Response: The order of references has been corrected.

  1. Statistical analyses must be inserted (figure 6, 7, 8, etc)

Response:  Statistical analysis is supplemented in Fig. 7 and 8.  Fig. 6 is used to calculate IC50, we added “error bars mean standard deviations from triplicated tests” in the revised manuscript.   

Reviewer 2 Report

In this study the authors showed that rhamnan sulfate obtained from Monostroma nitidum exerted strong inhibition of the interaction between heparin and SARS-CoV-2 spike protein and antiviral activities against some strains of SARS-CoV-2.

Line 34: Keywords – it is recommended that different keywords be used, which are not a part of the title.

Line 52-55, Line 84-86: The authors state that the RS may be a promising candidate in COVID-19 treatment or prevention. The RS is a sulfated polysaccharide with high molecular weight, that is, the compound may be difficult to get into the blood. The authors should describe how to use it to be effective as a therapeutic and/or preventive agent.

Line 64-72: Ref. 8-15 and Ref. 16, 19-22 – the citation is not valid. All the references should be validated.

Line 77: Based on our prior work (REF) – what does “REF” mean?

Line 169-170: The authors used pseudovirus but not SARS-CoV-2. Why was the “real” virus not used in the assays?

Line 195: the lowest dilution D at --- - what does “D” mean?

Figures 6 and 9: The RS was used at lower concentrations of ng/ml in Figure 6, while at much higher concentrations of mg/ml in Figure 9. Why did the authors do tests with a large concentration difference?

Figure 9A: The explanation of the images is not sufficient. These assays are qualitative, not quantitative. Why the authors did not evaluate the neutralizing activity of the RS by a plaque assay using SARS-CoV-2?

Author Response

 In this study the authors showed that rhamnan sulfate obtained from Monostroma nitidum exerted strong inhibition of the interaction between heparin and SARS-CoV-2 spike protein and antiviral activities against some strains of SARS-CoV-2.

Response: We thank reviewer #2 for the positive and helpful comments on our manuscript. 

  1. Line 34: Keywords – it is recommended that different keywords be used, which are not a part of the title

Response: The Keywords have been revised.

  1. Line 52-55, Line 84-86: The authors state that the RS may be a promising candidate in COVID-19 treatment or prevention. The RS is a sulfated polysaccharide with high molecular weight, that is, the compound may be difficult to get into the blood. The authors should describe how to use it to be effective as a therapeutic and/or preventive agent.

Response: Thanks for this great comment. It is true that RS is a polysaccharide with high molecular weight and is difficult to get into the blood.  We have envision two ways to use RS: i) reduce the MW of RS chemically or enzymatically to facilitate low molecular weight RS bioavailability; ii) COVID-19 results from initial SARS-Cov-2 contact with the nasal epithelium, binding of viral spike glycoprotein (SGP) to epithelial heparan sulfate delivering the virus to the ACE2 receptor, and then viral entry and infection. Formulation of RS as a nasal spray to block COVID-19 could be a potential therapeutic and/or preventative approach.

  1. Line 64-72: Ref. 8-15 and Ref. 16, 19-22 – the citation is not valid. All the references should be validated.

Response: All the references have been carefully checked and corrected.

  1. Line 77: Based on our prior work (REF) – what does “REF” mean?

Response: It was a mistake caused by reference management software. We’ve deleted “REF” in the revised manuscript.

  1. Line 169-170: The authors used pseudovirus but not SARS-CoV-2. Why was the “real” virus not used in the assays?

Response:  Using real SARS-CoV-2 in the assays requires biosafety level 3 (BSL3) facilities.  Pseudovirus has been widely used in the virology research community in biosafety level 2 (BSL2) labs. Our previous studies (ref. 6, 25, 26) demonstrated pseudotype SARS-CoV-2 virus particles working well in the in vitro antiviral assays.

  1. Line 195: the lowest dilution D at --- - what does “D” mean?

Response: The typo has been fixed.

Figures 6 and 9: The RS was used at lower concentrations of ng/ml in Figure 6, while at much higher concentrations of µg/ml in Figure 9. Why did the authors do tests with a large concentration difference?

Response:  In Figure 6, we used SPR to measure the inhibition activities of RS to S-protein-heparin interaction, which is in molecular level and needs lower concentrations of ng/ml. In the in vitro cell based assay (Figure 9), higher concentrations (µg/ml) were used due to the high molecular weight of RS.

  1. Figure 9A: The explanation of the images is not sufficient. These assays are qualitative, not quantitative. Why the authors did not evaluate the neutralizing activity of the RS by a plaque assay using SARS-CoV-2?

Response: The images provided are only a representation of a fluorescence measurement that corresponds to the infection of ACE-2 HEK293T cells by the SARS-COV-2 pseudoviruses. Briefly, pseudoviral particles were produced using a three-plasmid system, one of these coding for an EGFP reporter protein. Thus, if infection occurs, the reporter protein will be expressed, and cells will fluoresce. The IC50 value for the compound will be calculated through this measurement, as the prevention of the pseudovirus entry is higher with increasing concentration of RS, and thus, leading to lower EGFP expression.

The way EGFP expression is measured relies on the automated high-content screening (HCS) system, which used the Cellomics Arrayscan XTI (ThermoFischer Scientific). After the in vitro infectivity assay was conducted, cells were stained with Hoechst 33342 which enables the identification of individual nuclei, and thus single cells. Each well was then imaged using the HCS microscope at a 20X magnification using channels corresponding to blue and green fluorescence. Moreover, the lens used enables the collection of 64 images for each individual well within the 96-well plate, thus effectively discerning infected cells from non-infected cells for each replicate. The analysis software collects all this data and identifies individual nuclei (blue) as a region of interest (ROI). A second ROI for the EGFP channel is overlayed, thus enabling the calculation of a percentage of cells expressing EGFP, this corresponding to a percentage of infected cells.

The advantages of using this system rely on its high sensitivity to identify single cells. In the case of luminescent/absorbance detection systems, a plate reader would only be able to give an overall measurement from the entire well. On the other hand, a plaque-assay would only work for replicant-competent virus. The pseudovirus is a lentiviral vector where the Vsv-g gene is replaced by a gene for the spike protein. Such pseudoviruses are non-replicative, by design, and are used to provide infection of a cell, but cannot undergo replication within the cell. Thus, a standard plaque assay, as is used in live virus infectivity studies, cannot be performed. Nonetheless, with replicates and single-cell imaging, as well as relatively low multiplicity of infection, all as performed in this work, a highly quantitative measure of inhibition of pseudovirus infection of cells by RS could be performed.

Round 2

Reviewer 2 Report

When a polysaccharide with high MW reduce its MW, the biological activity is sometimes lost. The RS with lower MW should be evaluated for its activity. Nasal spray containing RS might be not safe, rather toxic.